# Radiofrequency Electromagnetic Fields Cause Non-Temperature-Induced Physical and Biological Effects in Cancer Cells

**DOI:** 10.3390/cancers14215349

**Published:** 2022-10-30

**Authors:** Peter Wust, Paraskevi D. Veltsista, Eva Oberacker, Prabhusrinivas Yavvari, Wolfgang Walther, Olof Bengtsson, Anja Sterner-Kock, Marie Weinhart, Florian Heyd, Patricia Grabowski, Sebastian Stintzing, Wolfgang Heinrich, Ulrike Stein, Pirus Ghadjar

**Affiliations:** 1Department of Radiation Oncology, Charité—Universitätsmedizin Berlin, Corporate Member of Freie Universität Berlin and Humboldt-Universität zu Berlin, Augustenburger Platz 1, 13353 Berlin, Germany; 2Institute of Chemistry and Biochemistry, Freie Universität Berlin, Takustr. 3, 14195 Berlin, Germany; 3Experimental and Clinical Research Center, Charité—Universitätsmedizin Berlin and Max-Delbrück-Center for Molecular Medicine, Germany and Experimental Pharmacology & Oncology (EPO), Robert-Rössle-Str. 10, 13125 Berlin, Germany; 4Max-Delbrück-Center for Molecular Medicine in the Helmholtz Association, AG Translational Oncology of Solid Tumors, Robert-Rössle-Str. 10, 13125 Berlin, Germany; 5German Cancer Consortium (DKTK), Im Neuenheimer Feld 280, 69120 Heidelberg, Germany; 6Ferdinand-Braun-Institut gGmbH, Leibniz-Institut Für Höchstfrequenztechnik, Gustav-Kirchhoff-Str. 4, 12489 Berlin, Germany; 7Experimental Pharmacology & Oncology GmbH Berlin-Buch, Robert-Rössle-Str. 10, 13125 Berlin, Germany; 8Institute of Physical Chemistry and Electrochemistry, Leibniz Universität Hannover, Callinstr. 3A, 30167 Hannover, Germany; 9Laboratory of RNA Biochemistry, Institute of Chemistry and Biochemistry, Freie Universität Berlin, Takustr. 6, 14195 Berlin, Germany; 10Department of Hematology, Oncology and Cancer Immunology (CCM), Charité—Universitätsmedizin Berlin, Corporate Member of Freie Universität Berlin and Humboldt-Universität zu Berlin, Charitéplatz 1, 10117 Berlin, Germany

**Keywords:** radiofrequency, amplitude modulation, hyperthermia, colorectal cancer, anticancer effects

## Abstract

**Simple Summary:**

Radiofrequency electromagnetic fields are used for tumor heating as adjunct therapy, but it appears that sufficient temperatures can sometimes not be reached. We therefore aimed to study potential non-temperature-induced anticancer effects when adding amplitude modulation to the radiofrequency waves. We could demonstrate in a colorectal cancer model that radiofrequency electromagnetic fields do have anticancer effects when not being induced by increased temperature that can be further increased by amplitude modulation. Therefore, this treatment could potentially serve as a more effective tumor therapy.

**Abstract:**

Non-temperature-induced effects of radiofrequency electromagnetic fields (RF) have been controversial for decades. Here, we established measurement techniques to prove their existence by investigating energy deposition in tumor cells under RF exposure and upon adding amplitude modulation (AM) (AMRF). Using a preclinical device LabEHY-200 with a novel in vitro applicator, we analyzed the power deposition and system parameters for five human colorectal cancer cell lines and measured the apoptosis rates in vitro and tumor growth inhibition in vivo in comparison to water bath heating. We showed enhanced anticancer effects of RF and AMRF in vitro and in vivo and verified the non-temperature-induced origin of the effects. Furthermore, apoptotic enhancement by AM was correlated with cell membrane stiffness. Our findings not only provide a strategy to significantly enhance non-temperature-induced anticancer cell effects in vitro and in vivo but also provide a perspective for a potentially more effective tumor therapy.

## 1. Introduction

Non-temperature-induced effects of radiofrequency electromagnetic fields (EMF) have been controversial for decades. Electromagnetic fields can be applied either as continuous waves (RF) or modified with additional amplitude modulation (AM), denoted as AMRF. In the literature, these effects are also, somewhat imprecisely, referred to as “nonthermal” effects. However, non-temperature-induced effects is a more suitable term, and it denotes the effects associated with the deposition of either thermal or nonthermal/isothermal energy. In the context of this work, we introduce the term of isothermal energy to denote energy absorption of a sample/tissue without resulting in a measurable temperature increase. We follow the thermodynamic term of an isothermal process, where a change in internal energy occurs without a concurrent temperature change. This is typically the case in slow processes where the rate of heat exchange is fast enough to prevent a temperature increase, for example, in cases in which the sample is in contact with a cooling reservoir. A non-temperature-induced effect is considered proven if RF or AMRF exposure leads to an increased physical or biological effect under otherwise identical conditions—in particular, identical temperature in the treated sample/region. However, the characteristics of the effects might change with the temperature.

RFs have been used in cancer therapy for decades under the label hyperthermia [1] to improve the effectiveness of chemotherapy [2] or radiation therapy [3,4]. According to the prevailing opinion, the therapeutic effect of RF was attributed only to the temperature increase induced by RF [1]. Due to the technical demands and limitations in achieving the required temperatures of >41 °C for tumor therapy [5,6], this therapeutic approach was only established for a few oncological indications.

Recently, non-temperature-induced antitumoral effects have been observed clinically. AMRF either at 13.56 MHz [7,8,9] or 27.12 MHz [10,11] were applied at rather low specific absorption rates (SAR) of 1 W/kg (or E-fields of 45 V/m) up to an estimated 15–20 W/kg (or E-fields up to 200 V/m) and temperatures below 40–41 °C or even at a normal body temperature. These AMRF techniques based on capacitive coupling of the EM field could be used with much less effort than the current clinical techniques centering around radiative EMF exposure and for significantly more indications in oncology.

Furthermore, preclinical studies showed various non-temperature-induced anticancer effects in vitro and in vivo applying AMRF at 13.56 MHz [12]. However, to the best of our knowledge, the precise contribution of AM was not investigated until now. This is considered a key question for our current study, because RF at 13.56 MHz (without AM) also showed non-temperature-induced antitumor effects when compared with water bath (WB) heating, both of which were carried out at 42 °C [13].

Calcium fluxes, which were observed in response to AMRF at 147 MHz, with low modulation frequencies around 16 Hz, were considered a key effect [14,15,16]. The various biological endpoints of electromagnetic radiation in the microwave range from mobile phones have also been investigated [17,18,19]. However, these investigations were mainly made on older cellular systems with pure phase-modulated signals and constant amplitude.

We believe that the long-standing apparent lack of clarification on the non-temperature-induced effects of harmonic RF or AMRF has two major reasons:So far, no reproducible measurement method has been established that can clearly detect differences in the power deposition in cells exposed either to RF or AMRF.There is a lack of irrefutable in vitro and in vivo experiments that compare WB heating with RF or AMRF and document additional RF and/or AMRF-induced anticancer effects. A typical knockout argument is the assertion of hidden thermal effects (local temperature rises, so-called hot spots). Therefore, such confounders must be methodically excluded using a thorough analysis of the experimental setup.

Using a series of physical measurements and in vitro and in vivo experiments, we addressed the above limitations and substantiated the non-temperature-induced anticancer effects of RF and AMRF. We also aimed to decipher the biophysical mechanisms underlying the antitumoral effects and to open up an innovative field for clinical application, particularly for treating cancer.

## 2. Materials and Methods

In this study we investigated the behavior of tumor cells in vitro and in vivo when exposed to RF and AMRF versus temperature alone (WB). We analyzed the physical parameters of the sample directly (measurement) or indirectly (system parameters), as well as the biological effectiveness of all treatments. A thorough investigation of the temperature distribution in a heterogeneous in vivo scenario was deciphered using EMF and temperature simulations.

### 2.1. Experimental System, Applicators, and Physical Parameters

We used the commercially available device LabEHY-200 (Oncotherm Kft., Budapest, Hungary) [20], as described in Figure 1, for RF and AMRF treatment with dedicated applicators for preclinical in vitro experiments with cell suspensions and in vivo experiments with mice tumors. The system consists of a signal generator with a fixed carrier frequency of 13.56 MHz (Figure 1A) and optional AM with a selectable pink noise spectrum with a noise density proportional to f_0_/f (for f ≥ f_0_ with f_0_ = 1, 10, 100 Hz, or 1 kHz). Modulation is applied with a selectable modulation index m (10–100%), where m defines the amplitude of the modulation wave with respect to the amplitude of the carrier wave (A_c_). In the frequency spectrum, the (AM) wave is no longer represented by a single peak at the carrier frequency (f_c_) but also exhibits two side peaks at the modulation frequency (f_c_ ± f_M_). Each of these peaks contributes to the total power carried by the EM wave:$$E={\;A}_{c}\cos\left(\omega_{c}t\right)+m\frac{A_{c}}{2}\cos\left(\left(\omega_{c}-\omega_{m}\right)t\right)+m\frac{A_{c}}{2}\cos\left(\left(\omega_{c}+\omega_{m}\right)t\right)$$

From this, the proportion of the AM side band power of the total signal power can be calculated as:

$\frac{m^{2}}{\left(2+m^{2}\right)}$, yielding 11% for m = 0.5 and 24% for m = 0.8

We used a novel in vitro applicator (Figure 1B) that allowed temperature regulation via temperature control loop operating a water flow cooling system, which was not possible using the previous applicator [13,21]. The in vivo applicator was used for small animal studies (Figure 1C) [22].

Each applicator can be interpreted as a series connection of capacitances (interfaces, deionized water, cell culture growth medium (Gibco Roswell Park Memorial Institute (RPMI) 1640 Medium, Thermo Fisher Scientific, Waltham, MA, USA), and tissues), the electrical parameters of which are listed in Table 1A. A capacitance of C = ε_0_ × ε_r_ × A/d was calculated for each single component (A: electrode/interface area, d: distance between electrodes/interfaces, ε_0_ = vacuum permittivity, and ε_r_ = relative permittivity of medium). A basic network analysis shows that the applicators have a total capacitance C ≅ 14 pF in the series, with an inductance L of ≅10 µH forming a series resonant circuit, which, in the lossless case, has the resonance frequency f:$$f=\frac{1}{2\pi \sqrt{\mathrm{LC}}}=13.56\;\mathrm{MHz}$$

Changes of the absorbed power in the sample represented by a loss resistance R in series with the lossless applicator (Figure 1E) change the resonance frequency. Any change of this absorbed power in the sample adds a slight frequency shift, resulting in detuning of the resonant circuit. A tuning circuit operated by a control voltage V and mechanically adjustable capacitors C_1_ and C_2_ (Figure 1E) compensates for a change in the absorbed power in order to achieve impedance matching and thereby maximum power transfer to the resonator, i.e., the applicator for the fixed carrier frequency 13.56 MHz. Practically, the control voltage V is tuned until the internally measured dissipated power P_diss_ (W) approaches the value of the adjusted power P_adj_ chosen for the treatment/experiment [22,23]. Typically, small differences <5% between P_adj_ and P_diss_ are achieved. The complex impedance **Z** of the applicator unit depends on L, C, and R and is given by the equation
$$Z=R+j\left(\omega L-\frac{1}{\omega C}\right)=Z\cdot e^{-j\theta }$$
with $\omega =\sqrt{\frac{1}{\mathrm{LC}}-\frac{R^{2}}{4L^{2}}}$, $Z=\sqrt{R^{2}+{\left(\omega L-\frac{1}{\omega C}\right)}^{2}}$, θ=arccot(ωL−1ωCR), showing the influence of R on the resonance frequency ω = 2πf of the circuit.

During measurements, Lab-EHY200 software automatically logs the internally measured values for the tuning capacitances C_1_ and C_2,_ power measurements (forward power P_forward_, reflected power P_refl_, and dissipated power P_diss_) and temperatures every second and saves to an output file. We analyzed the differences of the mean C_1_ and C_2_ between different cell lines and non-modulated RF versus AMRF, assuming that they are indicative of changes in the sample’s impedance, resulting in changes in the deposited power ΔP.

Determining the dissipated power quantified by the specific absorption rate (SAR in W/kg) in the sample is of particular interest and carried out as follows. At this point, it is important to note that the term of dissipated energy in the physical sense directly refers to energy absorption leading to a change in temperature, whereas the total absorbed power can contain isothermal energy in addition to dissipated energy. If the power is switched on at the initial thermal equilibrium, the SAR can be derived from the temperature gradient ΔT/Δt (in °C/min) quantifying the temperature increase ΔT during a given evaluated rising time Δt. The relationship between SAR and ΔT/Δt is derived from the bioheat transfer equation that is analytically solved for ∇^2^T = 0 (thermal equilibrium) [24,25]. The simple relationship:SAR [W/kg] = 60 × ΔT/Δt [°C/min](1)
is sufficient if the perfusions w (ml/100 g/min) and time intervals Δt are small enough (1st-order term). In our in vitro experiments, however, a high perfusion w must be assumed if the water flow is running in the in vitro applicator (Figure 1B). Then, the 2nd-order term is non-negligible, yielding:SAR [W/kg] = 60 × (1 + wρΔt/2) × ΔT/Δt [°C/min](2)

If we apply our evaluation interval of Δt = 60 s and a high w = 200 mL/100 g/min, this doubles SAR with Equation (2) (compared to Equation (1)). This has previously been confirmed by temperature–time measurements without the running water flow system at the same P_adj_ = 10 W that delivered about twice as high ΔT/Δt compared to water flow cooling ([26]). If we perform experiments with RF and AMRF under identical settings, a direct comparison of ΔT/Δt is possible after normalization to P_diss_ in order to compensate for slight inaccuracies in tuning < P_diss_ > to P_adj_: (ΔT/Δt)_corr_.

For each experiment, we determined and recorded changes of temperature gradients (ΔT/Δt)_corr_, as we compared AMRF to unmodulated RF.

For the in vitro studies, we applied a standard procedure to determine (ΔT/Δt)_corr_ from the respective datasets. We selected a reasonable fixed ramp-up time after the power is on for P_diss_ to approach P_adj_ (25 s). During the evaluation time Δt (60 s), we determined the slope of the regression line of the temperature–time curve. The length of Δt is a carefully considered tradeoff between statistics and linearity (Figure 1F). We considered typical fluctuations of the temperature–time curves with cycle times of 10–15 s to be related to the regulatory processes of the LabEHY-200 system.

For the in vivo studies, we had to carry out a graphically supported evaluation process (determination of the tangent) due to the variable settings and adjustments.

### 2.2. Applicator Models, Simulation Studies, and Isothermal Energy

In the case of the in vitro applicator, the temperatures in the cell suspension are assumed to be sufficiently homogeneous if the water flow is running, given the small sample size and thickness and the large contact area between the sample and the flowing water. The temperature is directly monitored in the cell chamber and further analyzed (Figure 1B).

On the contrary, we expect nonhomogeneous SAR and temperature distributions for the in vivo applicator (Figure 1C,D); therefore, we performed simulation studies for further clarification.

To determine the electric (E-) field, SAR, and temperature distributions, and particularly to exclude undesired thermal effects in the larger and temperature-destabilized volume of the mice treated in the in vivo applicator (Figure 1D), a realistic model of the applicator and a simplified model of a mouse were implemented in Sim4Life V6.0 (ZurichMedTech, Zurich, Switzerland).

For the E-field simulation using the quasistatic solver (13.56 MHz) of the FDTD (Finite Difference Time Domain) algorithm, we implemented the electrodes with the dimensions measured at the in vivo applicator (top electrode: ø = 1 cm, bottom electrode: 5.5 × 10.5 cm^2^). The top electrode was implemented as a solid structure due to the high conductivity of all the components, neglecting its composition of multiple thin rods held in place by a conductive textile to accommodate irregular surfaces. The voltage was directly applied to the top electrode, while the bottom electrode was considered grounded (Dirichlet boundary condition). After an initial simulation with U = 100 V, the voltage was adjusted until the absorbed power matched the adjusted power P_adj_ = 1 W chosen for the in vivo experiments. It is important to note that this power level refers to the continuous RF wave with constant amplitude, since implementing amplitude modulation is not supported in our simulation software. Given the power distribution between the carrier wave and the side bands of the modulation signal, this makes our simulation an upper bound approach to the problem. The mouse was approximated by a rectangle of 6 × 3 × 1.2 cm^3^, consisting of a 200 µm layer of fat enclosing homogeneous muscle tissue. The dimensions were both estimated from pictures taken during the in vivo experiments and adjusted to match the approximate weight of the mouse (here: m = 23 g) for a matching power balance. The overall resolution was set to 0.5 mm isotropic, with a high-resolution box (0.1 mm isotropic) centered underneath the electrode. For model A, a representation of the small tumor shortly after induction, a plane surface of the phantom was implemented. For model B, a sphere with a diameter of 8 mm placed at a depth of 3 mm of the rectangle (representing tumor growth) was used to bulge the surface of the phantom outwards by 1 mm. The superficial fat layer was maintained.

Note that, for in vivo experiments, a mixed tissue of muscle, fat, and intestine yields a mean heat capacitance c = 3600 Ws/kg/°C, and the relationship SAR (W/kg) = 60 × ΔT/Δt (°C/min) holds if the time interval Δt is sufficiently small (Equation (1)) [27].

If high SAR of >1000 W/kg are calculated in a small volume (approximated by a sphere with radius r (mm)), we can apply an analytical solution of the bioheat-transfer equation to quickly estimate the temperature increase [25]:ΔT_max_ [°C] = 4.2 × 10^−4^ × 4r^2^ × SAR, at the center(3)

According to this equation, even an excessive SAR of 10,000 W/kg focused to a sphere of 1 mm would result in a temperature increase of only 2 °C, which is not sufficient for any relevant thermal effect. However, accurate numerical calculations of temperature distributions are performed to confirm these estimations.

For the steady-state temperature simulation, the electric loss density result of the EMF simulation with P_diss_ = 1 W was used as the input. For this simulation, the full geometry, as shown in Figure 1D, was implemented to include the heat sink of the metal electrode rod. The initial body temperature and background temperature were set to 36 °C and 22 °C, respectively. All electrical and thermal properties are listed in Table 1. Temperature simulations were performed for model A. As a worst-case approximation, the perfusion in the tumor volume was set to zero. The temperature distributions depend considerably on the thermal conductivities λ of the electrode material and mouse tissue (W/m/°C), the heat transfer coefficient HTC (W/m^2^/°C), and the perfusion w (mL/100 g/min), according to previous studies [28,29]. The HTC for nude mice (6–10 W/m^2^/°C) was doubled in comparison to hirsute mice (Mus musculus, 3–5 W/m^2^/°C). We generated SAR volume histograms, 2D distributions, and cross-profiles of calculated SAR and temperature datasets. Finally, a comparison of the calculated and measured SAR (via a gradient of temperature increase ΔT/Δt) and the temperatures was performed.

If we measure different (ΔT/Δt)_corr_(AMRF) < (ΔT/Δt)_corr_(RF) after correction for P_diss_ (according to Figure 1F), such a difference suggests that isothermal power is deposited in the sample, which is caused by AM. Isothermal energy can primarily be stored electrostatically and chemically as quantified on a microscopic level by the equations in Figure 2 (left panel). A long-term macroscopic correlate for isothermal energy depositions are phase transitions (Figure 2, right panel), which we would expect particularly in cell membranes that can exhibit a transition from a solid to a fluid state at a certain temperature.

Under these conditions, an important aim of the present study was to evaluate the different physical parameters ((ΔT/Δt)_corr_, system parameters) for in vitro and in vivo studies, if either RF or AMRF is applied.

### 2.3. In Vitro Studies

We used five human colorectal cancer cell lines: HT29 (obtained from a primary tumor), SW480 (primary, Duke’s B), LoVo (left supraclavicular metastasis), SW620 (lymph node metastasis), and HCT116, which were originally purchased from the American Type Culture Collection (Manassas, VA, USA). The frozen cells were grown in RPMI 1640 medium and maintained at 37 °C in a humidified incubator with 5% CO_2_. We prepared cell suspensions with approximately one million cells per milliliter.

For the in vitro experiments, two milliliters of cell suspension were injected into the cell chamber of the in vitro applicator (Figure 1B). After careful venting, a fluoro-optic temperature sensor (FLUOROPTIC^®^ THERMOMETER m3300 Biomedical Lab Kit by Luxtron, ø = 0.5 mm) was positioned in the center via a provided channel under visual control. We always selected 42 °C as the target temperature for the flow system and an adjusted power P_adj_ of 10 W. In the case of AMRF, we selected 100/f (f ≥ 100 Hz) and a modulation index of 50%. This particular AM is also used in clinical systems, e.g., EHY-2000+ (Oncotherm Kft., Budapest, Hungary) and was utilized in two clinical studies [7,9]. The flow system runs a few minutes ahead of the power onset for thermal equilibration at room temperature.

For each of the five cell lines and the RPMI medium, 6 short experiments, i.e., 2 triplicates, were performed per arm (RF versus AMRF) to screen the physical data ((ΔT/Δt)_corr_, C_1_, C_2,_ t_meas_ = 10 min). Four cell lines (HT29, SW480, LoVo, and HCT116) were selected for further analysis that showed the highest differences in physical parameters between the medium and cell suspensions and between RF and AMRF exposition.

We hypothesized that AM with modulation frequencies of Hz to kHz (audio frequencies) can deposit energy in membranes by the excitation of mechanical vibrations [12]. The stiffness E (or Young’s modulus) (Pa) is a decisive parameter to estimate the fundamental (or ground) natural (or eigen) frequency f_M_ (Hz) of a membrane sheet with dimensions a × b (µm^2^) according to an equation derived in [30] (Vol. VI, §25):(4)$$f_{M}=120\;\times \;E^{1/2}\times (1/a^{2}+1/b^{2})$$

Therefore, we determined the stiffness E of normal cells (represented by hepatocytes and fibroblasts) in comparison to our selected tumor cell lines (HT29, SW480, LoVo, and HCT116) at different temperatures (37 °C and 41 °C) by indentation measurements using atomic force microscopy. The cells were probed using Bruker (Newark, DE, USA) MLCT-BIO-DC cantilevers with a spring constant of 0.01 N/m that were calibrated using a contact-based method on a JPK-NanoWizard^®^ 4 atomic force microscopy system. The cell culture dishes with the adherent cells were equilibrated at respective temperatures in HEPES buffer for 30 min prior to the experimentation, using the JPK PetriDishHeater^TM^ provided with the instrument. The actual temperature on the sample surface was checked with an IR thermometer on the sample surface prior to the measurements. The indentation experiments were performed at a setpoint of 1 nN and a relative setpoint of 0.8 nN to achieve an indentation depth of 500 nm, ensuring indentation at a larger surface area. A Z length of 4 µm with an extension speed of 1 µm/s and sample rate of 2046 Hz were used for collecting the data. For singularized, nonconfluent cells, three cells per sample were analyzed where five to six force curves were recorded for each cell in a 5 × 5 µm^2^ region. Each sample was analyzed in experimental triplicates. The analysis of the force curves was performed using JPK data processing software, and the Young’s modulus was calculated using the Hertz-Sneddon model for a triangular pyramid cantilever of 17° half-angle.

The three cell lines with the lowest stiffness E were selected for further in vitro analysis (HT29, SW480, and LoVo). We performed 4 triplicates per cell line and arm for the biological endpoint apoptosis (t_meas_ = 65 min: t_heat_ = 5 min + t_treat_ = 60 min). In parallel, controls were run in a water bath at 37 °C and 42 °C for the same time. Apoptosis rates were determined using FACS (fluorescence-activated cell sorting) by measuring the percentage of annexin V-positive cells. A direct comparison was performed by dividing the percentages of RF or AMRF by the mean percentage of the 42 °C water bath.

### 2.4. In Vivo Study

We used HT29 tumor s.c. xenografts for the in vivo study, because for HT29 cells, the largest differences between RF and AMRF had been observed in the in vitro experiments. The animal study with 6–8-week-old female NMRI nu/nu mice was performed at the Experimental Pharmacology & Oncology (EPO) GmbH Berlin-Buch (Berlin, Germany). All animals were maintained in a sterile environment with a daily 12-h light/12-h dark cycle, and sterilized food and water were provided ad libitum. Animals at 6–8 weeks old with a body weight of 22–25 g were used for xenografting. All experiments were carried out in accordance with the 3R (replace, reduce, and refine) and animal welfare (LAGESO Nummer 001/019), in accordance with the German law and Directive 2010/63/EU.

Subcutaneous injection at the femoral region with 1 × 10^7^ HT29 cells in 0.1 mL of phosphate-buffered solution (PBS) was performed to form tumor xenografts. Forty mice were randomly assigned to 4 groups of *n* = 10 animals. We compared the control group and whole body water bath hyperthermia group (WB) at 40 °C for 35 min with the RF group and AMRF group, for both adjusting a total power of P_adj_ = 1 W and a measurement time of 35 min (t_meas_ = 35 min: t_heat_ = 5 min + t_treat_ = 30 min). Direct temperature measurements were performed in the tumor center (as precisely as possible), at the skin in contact with the tumor, and in the rectum with the same temperature probes as used for the in vitro experiments. We determined the SAR after the onset of P_adj_ by applying tangents to the temperature time curves using a graphical method and registered P_diss_, C_1_, and C_2_ similar to the in vitro studies.

During the induction time of approximately 16 days (series 1), we started with the respective treatments on post-tumor inoculation days 7, 9, and 13. For this, the mice were anesthetized with isoflurane gas and received an injection of 150 mg/kg D-luciferin (Biosynth, Staad, Switzerland). We monitored the luciferase-expressing tumor xenograft activity by bioluminescence imaging using the NightOWL LB 981 imaging system (Berthold Technologies, Bad Wildbad, Germany). Images were analyzed by WinLight software (Berthold Technologies) and quantified using ImageJ 1.48v. In addition, tolerability and feasibility of the application were documented, i.e., potential skin irritations, treatment discontinuations, and the number of regular sessions. We measured the tumor volume (TV) by a digital caliper on study days 6 (baseline), 16, and 20.

On day 20 during the tumor growth period, we started a comparison of RF and AMRF in the so-far-untreated control group. We randomized 5 mice in each group and performed the three treatments on days 20 (after baseline measurements), 23, and 27. Here, we measured the TV on days 20, 23, 30, and 34 and again conducted bioluminescence imaging on days 20, 29, and 34 as the endpoints.

The anesthetized mice in the WB, RF, and AMRF groups were sacrificed on day 13 and in the control group on day 34 after the last TV measurement. The tumor samples were prepared for immunohistochemistry (IHC) by staining and evaluation for Ki67, because Ki67 protein (pKi67) expression is associated with the proliferative activity of intrinsic cell populations in tissues, including tumors. In evaluating the IHC results, the relative percentage of the immunopositive cells was assessed in relation to the total number of target cells. We defined a numerical Ki67 tumor score from 0 to 3.0 as follows: 0 (negative no stain), 0.5 (up to 12.5% of the cells show positive immunoreactivity), 1.0 (12.5–25%), 1.5 (25–37.5%), 2.0 (37.5–50%), 2.5 (50–75%), and 3.0 (75–100%).

### 2.5. Statistics

We used the software package GraphPad Prism v.6 for the statistics. Variance analyses between several series of measurements of two groups were performed by multiple *t*-tests and presented either as standard boxplots or dot plots additionally showing mean and mean error bars. Significance was defined as *p* < 0.05 and marked with * (<0.005 with **, etc.).

## 3. Results

### 3.1. Physical Parameters (In Vitro Applicator)

We compared RF at 13.56 MHz with AMRF, selecting the pink noise modulation frequency spectrum with a noise density ∝100/f with f ≥ 100 Hz and modulation index m ≅ 50%.

We evaluated and compared three physical parameters in five colorectal cancer cell lines prior to the in vitro and in vivo studies (Figure 3): temperature gradients (ΔT/Δt)_corr_ (°C/min) after power onset and system parameters C_1_ and C_2_.

Figure 3A shows that (ΔT/Δt)_corr_ under RF exposure were significantly higher by 15–20% for all cancer cell lines (HT29, SW480, LoVo, SW620, and HCT116) in comparison to the RPMI medium. When using AMRF, (ΔT/Δt)_corr_ in the medium was slightly (but non-significantly) higher (Figure 3A). If we, however, exposed cell suspensions to AMRF, (ΔT/Δt)_corr_ were significantly lower, close to the level found for RPMI medium. Given the normalization to the same power, these measurements suggest that cancer cells were subjected to isothermal energy if AM is added to RF, since the same amount of energy was absorbed, but a smaller fraction was dissipated as thermal energy.

The system parameters C_1_ and C_2_ (Figure 3B,C) also significantly differed between the medium and cell suspensions for both RF and AMRF exposure. C_1_ is significantly higher for SW480 cell suspensions when using AMRF compared to RF. C_2_ is more sensitive to a switch from RF to AMRF and is significantly lower for HT29 and SW480 cell suspensions.

### 3.2. Biological Effectiveness of RF and AMRF Versus WB In Vitro

We measured apoptosis, comparing exposure with RF and AMRF at a targeted temperature of 42 °C (measured mean of 41.8 °C) in the novel in vitro applicator to conventional water bath (WB) heating at the targeted 42 °C (measured mean of 42.3 °C) for the three selected colorectal cancer cell lines HT29, SW480, and LoVo. The measured temperature excess of 0.5 °C in the WB versus cell chamber (Figure 4A) confirms that additional apoptotic effects induced by RF or AMRF exposure cannot be caused by a higher temperature in the cell chamber and must be of non-temperature-induced origin.

For HT29 cells (Figure 4B), we found no relevant increase of apoptosis for RF versus WB (<10%) but significantly higher apoptosis for AMRF in comparison to either WB or RF (34% versus WB).

Contrarily, for SW480 cells (Figure 4C), we measured a significant increase of apoptosis for RF (45% versus WB); the apoptosis rates were enhanced at similar rates in both RF and AMRF. A similarly high anticancer effect of RF exposure on SW480 cells was previously reported [13]. For the cancer cell line LoVo (Figure 4D), we measured a slightly lower, but still significant, increase of apoptosis for RF (30% versus WB) and no further increase for AMRF. While (∆T/∆t)_corr_ under RF/AMRF behaves similarly for all three cell lines (Figure 3A), the lower sensitivity of LoVo to AM correlates with smaller differences in C_1_ and C_2_ when AM is used (Figure 3B,C).

To further decipher the reason behind these varied responses of the cell lines to AM with f_0_ = 100 Hz, we measured the stiffness E (or Young’s modulus) of hepatocytes and fibroblasts (representative for normal tissue cells) and the selected cancer cells as a function of the temperature (Figure 4E, left panel) using atomic force microscopy. The stiffness of normal cells at 37 °C ranges at 10–25 kPa, while the stiffness of cancer cells is much lower but still in the range of 2–3 kPa. However, a dramatic drop of E is observed in the cells when the temperature is increased to 41 °C. In the cancer cells, E declines to 200–600 Pa, which corresponds to lower mechanical resonance frequencies f_M_, which now fall into the audio frequency range (explained in Methods, Equation (4)).

A more detailed analysis of the cancer cells at 41 °C revealed significant differences of E between the cells and the derived membrane resonances (Figure 4E, right panel). The metastatic, i.e., aggressive cancer cells LoVo/HCT116 have the lowest E of approximately 200 Pa, as opposed to 600 Pa for HT29 cells. If we assume a spherical segment with an extension of 7 µm as the largest possible vibrating membrane section of a spherical cell with a 10 µm diameter, we obtain the fundamental characteristic resonance frequencies f_M_ of each cell according to Equation (4). A membrane resonance frequency above 100 Hz is predicted for the HT29 cells, which is thus covered by the applied AM spectrum 100/f (f ≥ 100 Hz). For the other cell lines, we obtain fundamental membrane frequencies below 100 Hz. Thus, a stimulation of fundamental membrane resonances using this AM spectrum, which might cause membrane damages, is only expected in HT29 cells (and not in LoVo and SW480 cells). This is in agreement with our in vitro data (Figure 4B,D).

### 3.3. Biological Effectiveness of RF and AMRF Versus WB In Vivo

The in vitro data prompted us to perform an in vivo study with the luciferase reporter gene stably expressing HT29 s.c. xenografts using the same AM spectrum.

In series 1, we assessed 4 groups of *n* = 10 mice each, comparing the relative bioluminescence signal at the start of treatment on day 1 and after 3 treatments (running until day 5) on day 12 (Figure 5A). We measured a slight but nonsignificant increase in the control group and WB group (heating at 40 °C, t = 30 min) and a slight nonsignificant decrease in the experimental RF group (RF, P_adj_ = 1 W, t = 35 min, T_max_ < 40 °C); however, the experimental AMRF group showed a significant drop (same settings as the RF group). Note that the in vivo treatments were distinctly less intensive compared to the in vitro studies (treatment times 35 min in vivo versus 65 min in vitro and T_max_ < 40 °C versus T_mean_ ≅ 41.8 °C in vitro (Figure 4A)). Nevertheless, the effects in the in vivo AMRF group were significant regarding the antitumoral efficacy.

Figure 5B shows bioluminescence images with typical examples of progression (control group), stable disease/minor response (WB and RF group), and significant antitumoral response, reflected by tumor reduction (AMRF group).

The scores for proliferation marker Ki67 also showed a significant decrease for the AMRF group compared to the WB group (Figure 5C, left panel), while the RF group showed at least a tendency to be more effective than WB heating. Figure 5D shows the highest Ki67 tumor score 3.0 and a rather low Ki67 score 0.5 for illustration (for scoring, see Methods). Additional AM clearly causes a significant improvement of anticancer effectiveness.

Since series 1 (up to 20 days after implantation, Figure 5B) fell mainly in the induction phase (up to 16 days after implantation), tumor volume measurements were not performed on the small tumors (with measured diameters 6 mm).

For the untreated control group of series 1 (*n* = 10 mice), we conducted a second treatment series during the tumor growth period, with larger tumors (diameter 8 mm) starting on day 20 after implantation. We randomized the mice into an RF and AMRF group of *n =* 5 mice each treated on days 20, 23, and 27 either with RF or AMRF at the same settings as used for series 1. We again found significantly lower Ki67 scores (Figure 5C, right panel) and a tendency to lower the relative luminescence signals (Figure 5E) in the AMRF group. In particular, the tumor growth curve of the AMRF-group was significantly flattened compared to the RF group (Figure 5F), showing the higher effectiveness of the AMRF treatments. Furthermore, we found less progressive disease (40% versus 80%) and more stable disease (60% versus 20%) in the AMRF arm (Figure 5F).

Pathological examination of the tumor tissues revealed no indication for treatment-related tissue damage or necrosis in the surrounding healthy tissues.

### 3.4. Physical Parameters (In Vivo Applicator)

We evaluated C_1_, C_2_, (ΔT/Δt)_corr_ for HT29 xenografts treated with the in vivo applicator and found the same significant differences between RF and AMRF as for the in vitro applicator comparing all RF and AMRF measurements (Figure 6A,B).

We also analyzed (ΔT/Δt)_corr_ as a function of the measurement location (either superficial in tumor contact or in the implanted intratumoral catheter) (Figure 6B, right panel) and determined the mean maximum temperatures in the same measurement points (Figure 6C).

For the first in vivo study (Series 1: RF versus AMRF group, small tumors of 6 mm in diameter), the thermally relevant (ΔT/Δt)_corr_ values (after power onset) tend to be higher at the surface for the RF group, but the opposite was observed for the AMRF group (Figure 6B, middle). The values of the difference (ΔT/Δt)_skin_ − (ΔT/Δt)_deep_ were significantly different when comparing RF to AMRF. Such nonhomogeneous temperature distributions (i.e., big differences between superficial and deep measurement points) are considered unfavorable indicators both in terms of toxicity and effectiveness.

For the second in vivo study (Series 2: RF versus AMRF in the former control group) with larger tumors of 8 mm in diameter in the growth phase, we again found significantly lower values of (ΔT/Δt)_corr_ in the AMRF group, which were consistent with our findings in the in vitro applicator.

Although we measured lower thermally relevant dissipated power for AMRF exposure (e.g., 40% reduction for Series 2), the final mean temperatures <T_max_> in the steady state were not reduced and, in particular, were more homogeneously distributed, i.e., showed a smaller difference (ΔT/Δt)_skin_ − (ΔT/Δt)_deep_ (compare the RF and AMRF groups, Figure 6C, left panel).

Figure 6D shows that the feasibility and tolerance of the AMRF treatments were significantly greater compared to the RF treatments, with 33% more treatments being completed according to protocol (i.e., without complications such as skin irritations) under the same conditions for AMRF. This could be explained by the lower thermal power densities (Figure 6A,B) and more homogeneous temperature distributions of AMRF (Figure 6C).

Both the in vitro and in vivo data consistently show that, for the same applied power, the thermally relevant dissipated power is lower for AMRF, indicating that portions of the absorbed power are transformed into isothermal energy. The in vivo data show, in addition, that this transformation increases the tolerance and is associated with more homogeneous temperature distributions.

In summary, the in vivo study confirmed the in vitro data both physically and biologically. Moreover, the AM-related effects were even more prominent in vivo.

### 3.5. Measured and Simulated Temperatures for the Applicators

The temperatures measured in the in vitro applicator are representative for the entire cell chamber, because this small volume of 2 mL with a 3 mm thickness is entirely in close contact with a highly effective water flow system for temperature regulation.

The conditions are more complicated for the in vivo applicator and prompted us to carry out SAR and temperature simulations. Our calculations yielded strongly nonhomogeneous SAR distributions for two models; A and B (Figure 7A,B upper/middle row), corresponding to the two tumor growth stages of Series 1 and 2. Outside the tumors, applicator edge effects generated SAR maxima (see profiles at depth d = 0.05 mm) that might cause skin irritations at these locations (Figure 6B). However, the SAR volume histograms of the tumors (of either 6 or 8 mm in diameter) revealed that the volumes in the tumors with excessive SAR (up to 6500 W/kg) are too small (<0.1 mL) to induce a significant localized temperature increase, as estimated by the 42 °C curves (see Methods, Equation (3)), resulting in intratumoral temperatures far below 42 °C (Figure 7A,B bottom). Even when considering the temporal peak power levels of the modulated wave (up to 150% for m = 0.5), the peak SAR values will not induce a temperature increase of 2 °C or more (see Methods, Equation (3)). The calculated SAR_50_ in the tumors are consistent with the mean values of our measurements (see inlay, Figure 7A,B, third row).

The simulated temperature distributions (Figure 7C) confirmed that the intratumoral temperatures barely reach 40 °C, in agreement with the direct tumor-related temperature measurements, but indicate the possibility of overheating at the applicator edge outside the tumor.

Our careful theoretical analysis of the in vivo applicator could rule out hot spots (temperatures > 40 °C) in the implanted tumors.

## 4. Discussion

We identified non-temperature-induced anticancer effects due to RF in three colorectal cancer cell lines in vitro, which were increased by AMRF in one of the cell lines in vitro, using a frequency spectrum for modulation (f ≥ 100 Hz, 100/f spectrum). This could be confirmed in vivo, bearing the potential for rapid clinical translation.

Our measurements of the physical parameters revealed that colorectal cancer cells absorb up to 20% additional thermal power from a harmonic RF at 13.56 MHz compared to WB heating. This might be explained by the change of the tissue interactions with EM waves in the investigated frequency range. The delta dispersion on the tail of a beta dispersion [31] around 10 MHz is characterized by a change from membrane-dominated interactions in the β region to dipolar interactions of bound water or side chains of molecules. Owing to the interaction of RF with proteins and protein-bound water, the delta dispersion increases the value of the dielectric constant ε_r_ up to several hundred, which can result in increased interaction of the RF with the cancer cells [32]. Moreover, others have previously described the mitochondria as a possible specific target of RF at 13 MHz [33,34]. Wust et al. [13] described that a slight DC voltage at ion channels, generated from the RF by rectification and smoothing, can lead to ion disequilibrium of cells—in particular, for Ca^2+^ ions.

If we use AMRF, this excess of dissipated power from RF is decreasing and presumably converted into isothermal energy during the heating-up phase. Particularly, phase transitions—for example, the transition of the cell membrane from a solid-like to a liquid-like state, eventually leading to cell damage—consume substantial isothermal energy.

A demodulation of the AMRF results in the low modulation frequency acting upon the cell membrane and the ion channels thereon [12]. This can trigger a mechanical vibration of membrane parts with a frequency covered by the modulation frequency spectrum, leading to, for instance, a decrease of the membrane potential, opening of voltage-gated channels, or triggering of other processes [12]. Such processes consume isothermal energy but are not necessarily cytotoxic.

A mechanically induced cytotoxicity is expected if a mechanical ground resonance of the whole cell is excited, which we considered as the vibration of a spherical segment as large as possible (estimated as 7 µm diameter for a cell of 10 µm in diameter). We measured the cell membrane stiffness E of the used cell lines by atomic force microscopy and used the results to calculate the potential resonance frequencies of the cells. Our results show that, only for the cell line in which AMRF outperformed the anticancer effects of RF in vitro, the calculated resonance frequency was above 100 Hz, thus being included in the frequency spectrum used for modulation. As this explains the differences between the cell lines, it supports our hypothesis.

Since the interactions at the cell membranes are considered as pure E-field effects [12], the product of E-field (V/m) × time (min) can reasonably be considered as a dose to predict non-temperature-induced antitumoral effects in clinical applications. For our experiments, we estimate a dose of 60,000 V∙min/m per in vitro experiment (of 60 min), while the mean dose per in vivo treatment (of 30 min) was half this value. Since we performed three treatments per arm (see Methods), we achieved a total dose of 90,000 V∙min/m for each in vivo experiment.

These preclinical dosages can be compared with clinically achievable doses. An SAR of 20 W/kg (E = 200 V/m) is realistic in the pelvis and abdomen with capacitive systems [35]. Therefore, a dose of 12,000 V∙min/m can be estimated for a clinical treatment of 60 min. If we consider a typical scheme of 10 treatments (2 treatments per week), we achieve even higher doses of 120,000 V∙min/m for a full course (24,000 V∙min/m per week). Thus, a clinical translation of our preclinical results appears feasible.

Even treatments with comparatively low levels of 1 W/kg (E = 45 V/m) for 3 h (on 3 days per week) deliver similar doses of 24,300 V∙min/m per week [36,37].

The existence of extraordinary modulation frequencies has been proposed by others in the past [10,11,14,15,36,37,38] and might be the subject of future studies. In addition, further studies are required regarding the required number, duration, and interval of treatments to optimize the non-temperature-dependent anticancer effects.

## 5. Conclusions

Our study verified the existence of non-temperature-induced anticancer effects of RF in three tested cell lines in vitro. Moreover, we could demonstrate a further increase of the non-temperature-dependent anticancer effects due to AMRF in one tested cell line in vitro and in vivo. RF/AMRF promises a locoregional oncological therapy using both temperature effects (hyperthermia) and non-temperature-induced anticancer effects with the potential to serve as a more effective tumor therapy. More research in terms of the choice of amplitude modulation frequency and choice of carrier frequency among other variables is required.

## Figures and Tables

**Figure 1 cancers-14-05349-f001:**
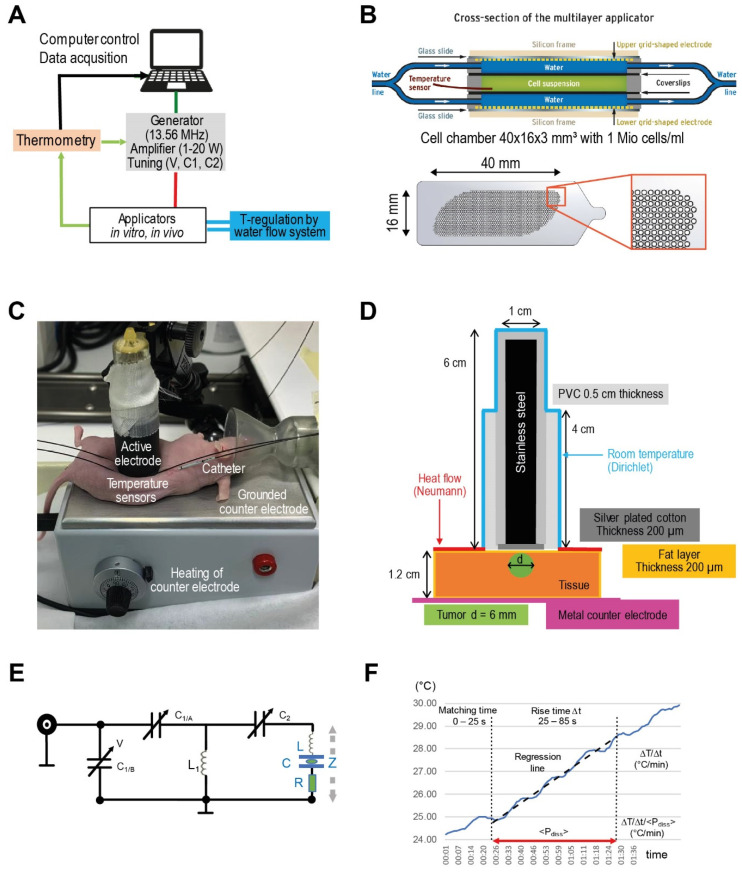
(**A**) Experimental system. Schematic of LabEHY-200 system for preclinical in vitro and in vivo studies. With reference to the clinical system EHY-2000+ [8], we initially set the AM-spectrum to 100/f (f ≥ 100 Hz). (**B**) In vitro applicator. Schematic cross-section of the in vitro applicator, which is suitable to adjust the power density (via P_adj_) and temperature (via water flow system) separately. The grid-shaped electrodes (below) enabled enlargement of the cell chamber volume (green) up to ~2 mL. (**C**) In vivo applicator. Components of the in vivo applicator in situ as positioned upon an implanted HT29 tumor in a nude mouse. (**D**) Model of in vivo applicator. The exposed volume is much smaller than for the in vitro applicator, resulting in a P_adj_ = 1 W. Particularly, calculating temperature distributions is a nontrivial task because of this complicated geometry. (**E**) Tuning circuit. Tuning circuit for the in vitro applicator with adjustable capacitances C_1_/C_2_, which are controlled by the voltage V. The complex impedance **Z** is specified by the fixed inductance L (air coil) in a pre-tuning unit, the capacitance C and a variable lossy resistance R, depending on the load and operation mode. (**F**) Evaluation procedure for the temperature gradient after the power is turned on. Standardized evaluation of ΔT/Δt (°C/min) in the in vitro applicator after switching on the power (P_adj_ = 10 W) under flow conditions in three steps (1–3): (1): Fixed tuning time of 25 s to achieve <P_diss_> = P_adj_ with a reflected power of <5%. (2): Determination of the slope ΔT/Δt of the regression line in the time interval 25–85 s (red line). (3): Correction of the temperature gradient with respect to the mean P_diss_ during this time interval.

**Figure 2 cancers-14-05349-f002:**
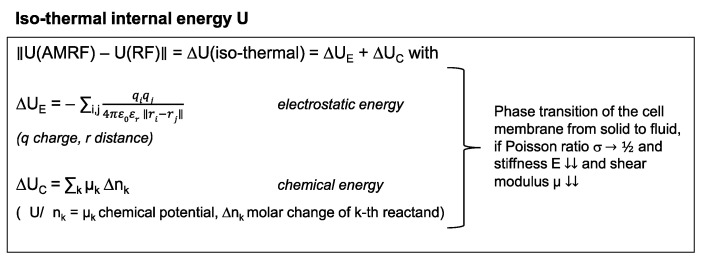
Isothermal energy. Isothermal contributions to the internal energy of tumor cells. These energy depositions do not cause a temperature increase, especially during the heat-up phase. In particular, phase transitions substantially consume isothermal energy. Such phase transitions might occur at the cell membrane under certain circumstances.

**Figure 3 cancers-14-05349-f003:**
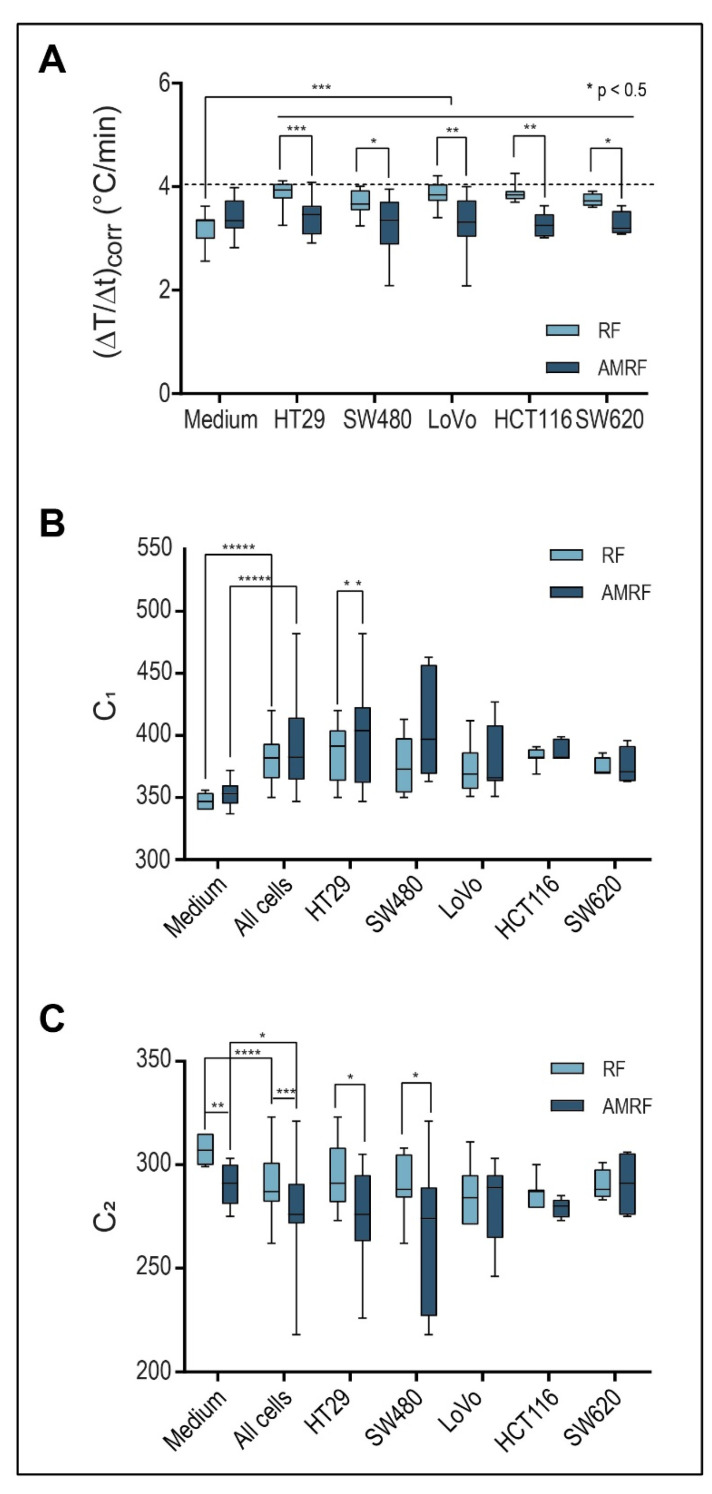
(**A**) Power deposition; Detected temperature gradients after radiation onset in the in vitro applicator (10 W with flow, ± AM with ≥ 100 Hz), which are significantly higher for RF in all cell suspensions than in the medium. For RPMI medium, a slight nonsignificant increase is observed for AMRF versus RF. In contrast, for each cell line, a significantly lower value is observed (significance level is shown above the bars: *p* < 0.05 and marked with *, < 0.005 with **, etc.). The dashed line marks the medium level for RF. (**B**) Tuning capacitance C_1_; C_1_ of the in vitro applicator for medium and five cell lines exposed to RF or AMRF radiation. Differences are evident between medium and cell suspensions and between RF and AMRF—in particular, after pooling the cells. Individual cell lines appear to follow a trend. C_1_ is higher when using AMRF instead of RF. The difference is lowest for the cell lines (LoVo, SW620, and HCT116). (**C**) Tuning capacitance C_2_; C_2_ of the in vitro applicator for medium and five cell lines exposed to RF or AMRF radiation. Differences occur similar to C_1_, but C_2_ values are lower for AMRF than for RF. Again, this difference is lowest for the cell lines (LoVo, SW620, and HCT116).

**Figure 4 cancers-14-05349-f004:**
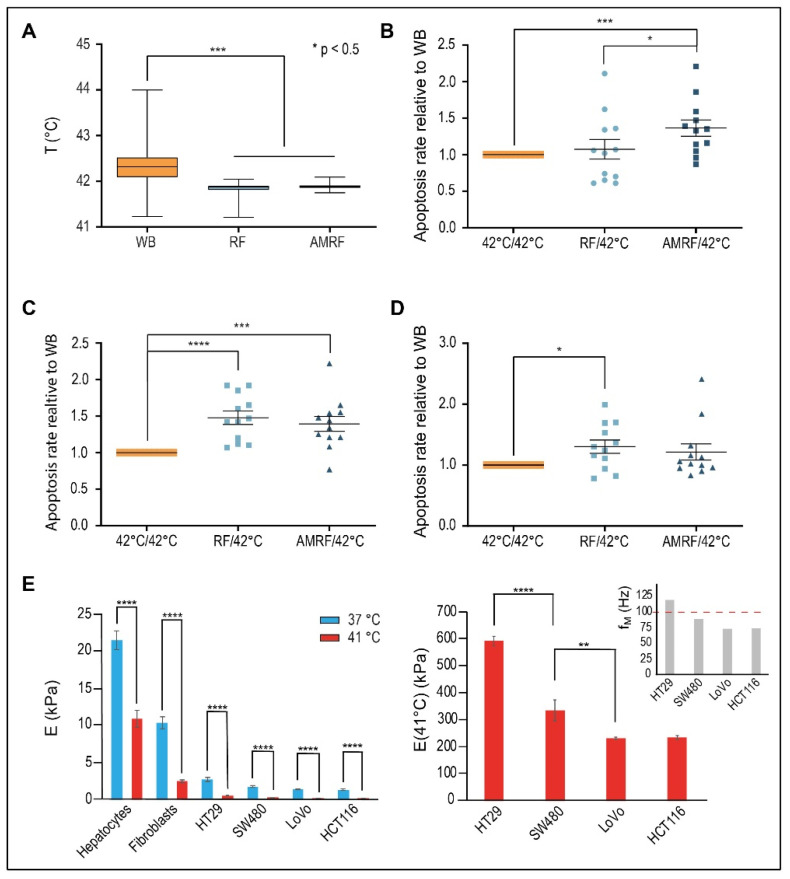
(**A**) Measured temperatures in vitro. Mean temperatures during the 60-min treatment time measured in the cell chamber for RF or AMRF in comparison to the water bath (WB). A sample of 26 biological experiments enclosing all cell lines was evaluated. (**B**) Normalized apoptosis rate in vitro for HT29 cells. Evaluation of apoptosis rates (annexin V+) for HT29 cells in suspension in the in vitro applicator comparing RF and AMRF versus those of cells in a water bath adjusted to 42 °C by normalizing to the mean apoptosis rate of the WB experiments. Significantly higher apoptosis rates are detected in the AMRF arm. Estimated ground natural frequency of the membrane f_M_ is covered by the AM spectrum (see (**E**)). (**C**) Normalized apoptosis rate in vitro for SW480 cells. Apoptosis rates are equally enhanced in the RF and AMRF arms. Estimated ground natural frequency of the membrane f_M_ is slightly below the AM spectrum (see (**E**)). (**D**) Normalized apoptosis rate in vitro for LoVo cells. Apoptosis rates are equally enhanced in the RF and AMRF arms. Estimated ground natural frequency of the membrane f_M_ is clearly below the AM spectrum (see (**E**)). (**E**) Young´s moduli E for different cell lines and temperatures, as determined by atomic force microscopy indentation. The overview (left panel) shows that normal issue cells (hepatocytes and fibroblasts) have higher E than cancer cells. A significant drop of E occurs for each cell type as the temperature increases from 37 °C to 41 °C. The more detailed representation (right panel) shows that the moduli at 41 °C significantly differ between cancer cells and lead to different resonance frequencies f_M_, which are below or above 100 Hz, according to Equation (4) (Section 2 Methods): f_M_ = 120 × E^1/2^ × (1/a^2^ + 1/b^2^). Significance was defined as *p* < 0.05 and marked with * (< 0.005 with **, etc.)

**Figure 5 cancers-14-05349-f005:**
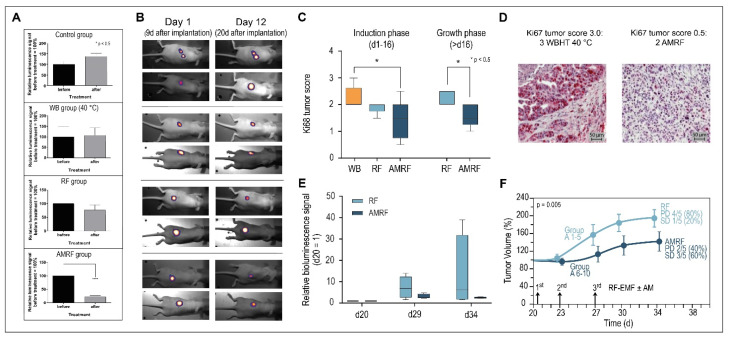
(**A**) Relative bioluminescence signals (series 1). Response after three treatments (measured on day 12 after treatment) for all groups, showing the significant drop for the AMRF group. (**B**) Bioluminescence images. Demonstration of progression (control group), stable course (WB and RF groups), and response (AMRF group). (**C**) Immunohistochemistry. Expression of Ki67 antigen after 3 WBHT or RF/AMRF treatments on day 13 (series 1, left panel) or after 3 RF/AMRF on day 34 (series 2, right panel). The decline of Ki67 after 3 AMRF is significant. (**D**) Immunohistochemical staining. Representative images of Ki67 immunohistochemical staining, with the highest score 3.0 (top) and a low score 0.5. (bottom). (**E**) Relative bioluminescence signals (series 2). Response after three treatments on d20, d23, and d27 in the RF and AMRF groups as randomized in the control group, normalized to the signal on d20. Due to the large scatter in the RF group, no statistical significance was achieved. (**F**) Tumor growth curves (series 2). Significantly different tumor growth curves for the RF group versus AMRF group after three treatments on days 21, 23, and 27. Progressive disease (PD) is halved in the AMRF group (40% versus 80%) and stable disease (SD) tripled (60% versus 20%). PD: tumor volume > 150% on d34. SD: tumor volume 50–150% on d34. Significance was defined as *p* < 0.05 and marked with * (< 0.005 with **, etc.)

**Figure 6 cancers-14-05349-f006:**
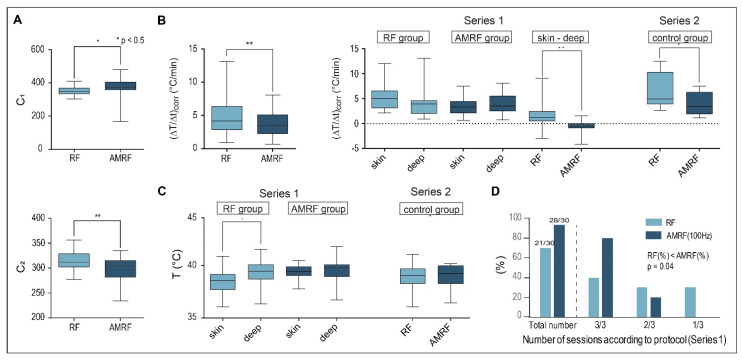
(**A**) Tuning capacitances in vivo. Values obtained for C_1_ and C_2_ in vivo show the same qualitative behavior as those achieved in vitro (Figure 1B,C), with even higher statistical significance. We included all completed RF and AMRF treatments in this analysis. (**B**) Corrected power deposition in vivo. The initial temperature gradients (ΔT/Δt)_corr_ (corrected for the dissipated power) in vivo behave similar to the in vitro measurements (Figure 3A) if all RF and AMRF measurement points were compared. The temperature gradients as a function of the locations (superficial versus deep in the tumor center) and techniques (RF versus AMRF) suggest that the thermally relevant power density is higher and more nonhomogeneous for RF in comparison to AMRF. (**C**) Location-dependent temperatures in vivo. Mean T_max_ in tumor-related superficial and intratumoral deep measurement points broken down by RF (light) and AMRF (dark) for series 1 (left panel) and series 2 (right panel). Even though the thermal power is less for AMRF, the measured temperatures are comparable and more homogeneous. (**D**) Feasibility and tolerance. Three sessions per mouse were intended in series 1 (RF group versus AMRF group with 2 × 10 mice). The number of successfully completed sessions is significantly higher in the AMRF group. Significance was defined as *p* < 0.05 and marked with * (< 0.005 with **, etc.)

**Figure 7 cancers-14-05349-f007:**
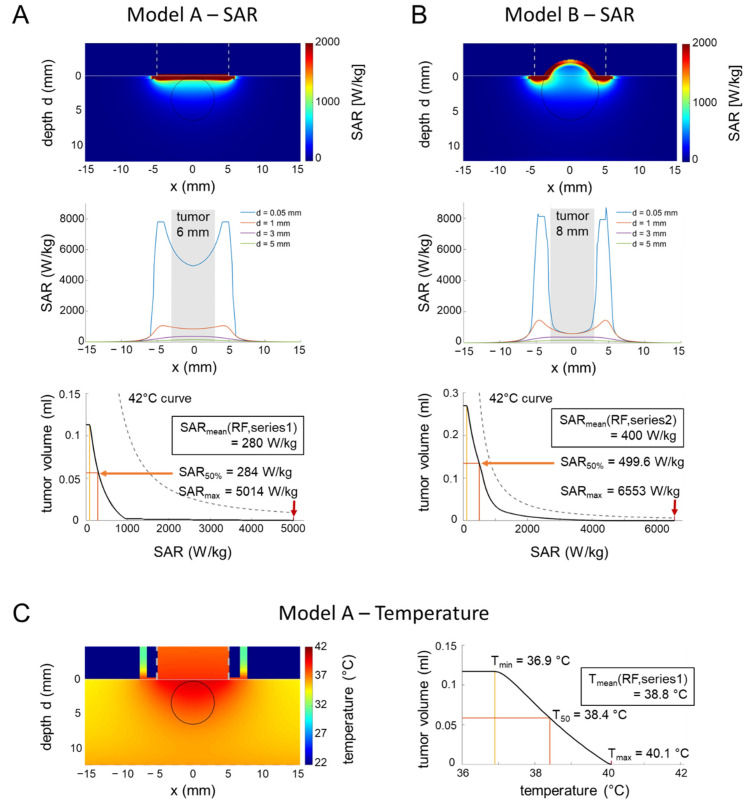
(**A**) SAR simulations in an in vivo applicator (small tumor). Two-dimensional SAR distribution (top), depth profiles near the surface (middle), and SAR volume histogram of the tumor for model A representing series 1 with small tumors (6 mm in diameter) during the induction phase. The SAR volume histogram of the tumor is far below the 42 °C curve. SAR_min_, SAR_50%_ and SAR_max_ are marked with yellow, orange and red markers respectively. However, overheating might be possible outside the tumor at the applicator edge under unfavorable conditions (see profile at depth d = 0.05 mm). (**B**) SAR simulations in an in vivo applicator (large tumor). For model B with larger tumors (8 mm in diameter), the SAR volume histogram yields higher SAR than for model A but is still below the 42 °C curve (please note the adapted scale of the y-axis). Similar to model A, overheating might be possible outside the tumor at the applicator edge with similar SAR peaks (see the profile at depth d = 0.05 mm). (**C**) Simulation of temperature distributions. The temperature distribution for model A with the applicator model of Figure 7B and the thermal parameters of Table 1B confirm the estimations of the SAR analysis. The temperature volume histogram of the tumor shows a maximum temperature of 40.1 °C.

**Table 1 cancers-14-05349-t001:** (**A**) Electrical parameters (electrode size A, relative permittivity ε_r_, and electric conductivity σ) required to calculate the capacitances C in vitro and in vivo and to simulate E-field distributions. (**B**) List of thermal parameters required to calculate the temperature distributions for the in vivo applicator, which is applied for nude mice.

**(A)**			
**Electrical Parameters**	**In Vitro Applicator**		**In Vivo Applicator**
electrodes upper electrode counter electrode	grid-shaped electrodes (steel) 4 × 1.6 × 0.75 cm^2^ electrode area reduced by 0.75		stainless steel ø _1_ = 1 cm, h = 6 cm ø _2_ = (ø _1_^2^ + d^2^)^1/2^
distance d (cm) between electrodes	1.3		1.2 (mouse)
RPMI-Medium between electrodes	ε_r_ = 72.5, σ = 1.2 S/m		ε_r_ = 100 (2/3 medium) ε_r_ = 79 (phantom)
additional interfaces	coverslips, polyolefin 2 × 0.2 mm, ε_r_ = 2		silver plated cotton 0.2 mm, ε_r_ = 2
**(B)**			
**Thermal Parameters**		**In Vivo Applicator**	
Thermal conductivity λ (W/m/°C)		Electrode (steel): 20 Cover (silver plated cotton): 90 Tissue (mixture muscle/fat): 0.5	
Heat capacitance c (Ws/kg/°C)		Electrode (steel): 420 Tissue: 3600	
Heat transfer HTC (W/m^2^/°C)		Mouse body: 6	
Perfusion (mL/100 g/min)		Tissue: 100	
Heat generation (W/kg)		Tissue: 10	

## Data Availability

The data can be shared up on request.

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
