# Peer review of "Radiofrequency Electromagnetic Fields Cause Non-Temperature-Induced Physical and Biological Effects in Cancer Cells"

_cancers, 2022, doi:10.3390/cancers14215349_

Round 1

Reviewer 1 Report

The submission is a gap-filling study on the emerging field of hyperthermia in oncology, solving the challenge of modulation of radiofrequency (RF) heating. Many modulated RF applications show remarkable medical successes but do not show the additional effect of modulation on unmodulated RF heating. The present submission experimentally compares the modulated and unmodulated effects in vitro and in vivo. To my knowledge, this novel work is the first study of this kind. This systematic work examines the subject from various angles. The scientifically well-communicated results could significantly step forward in improving modern oncological hyperthermia.

However, before publishing, it needs numerous minor corrections. I give my comments in the order of the rows, indicating their numbers.

Comments:

92. The mobile phones are microwaves and use phase-modulation, which is not comparable with amplitude modulation of RF.

116. This applied modulation is not pink noise. In the pink noise, the noise-power-density () is proportional to .

119. Give the reference for the formula.

126. Correct  to  (relative dielectric constant).

145. The correct line is:
; ;

150. (Table 1A.) The meaning of the used  and  (dielectric constant and conductivity) have to be explained. The RPMI medium explanation must also be here (it is explained only later in row 273). In the last row of Table 1.  is needed instead of  (two times).

157. The change of deposited power is . (Note, the change of  and  may indicate the change of the real part , and the imaginary part  of the impedance.)

160. A description is needed, what is denoted with . Its definition is also missing in Figure 1. (row 385).

237. The study correctly uses NMRI nu/nu nude mice (immunodeficient mice) to avoid the rejection response of human cell lines. What is the comparison to the mentioned “laboratory mice”?

240. The time interval is .

243. The notation  was used in the definition of the capacity  ( in row 126.). The new notation here would be more precise.

292. The article does not show the results of  water bath.

300. The mechanical vibration has a vital role in cellular degradation, but only this can not produce selected production of proteins like membrane enriched calreticulin [[i]], extracellular HMGB1 [[ii]], E2F1-mediated apoptosis [[iii]], the caspase independent intrinsic pathways of apoptosis starting from mitochondria by AIF [[iv]] and the  influx [[v]] are also not explainable.

340. The notation PBS has to be named (phosphate buffered saline). 

342. The in vivo method of water-bath heating is missing. Was it whole-body treatment?

358. What had signalized the skin irritation?

460. Typing mistake. The correct is: .

486. The membrane damage is only one part of the effect. The induction of immunogenic processes is more than regular damage [[vi]]. 

538. The size of the intratumoral temperature sensor (optical) has to be mentioned (here or earlier in the text).

555. It is not completely clear why the depth effect of hyperthermia is unfavorable.

697. Typing mistake:  is requested.

After the corrections, I strongly support the publication of this essential contribution to solving the challenges of modern hyperthermia in oncology.

[[i]]          Yang K-L, Huang C-C, Chi M-S, Chiang H-C, Wang Y-S, Andocs G, et.al. (2016) In vitro comparison of conventional hyperthermia and modulated electro-hyperthermia, Oncotarget, 7(51): 84082-84092, doi: 10.18632/oncotarget.11444, http://www.ncbi.nlm.nih.gov/pubmed/27556507

[[ii]]         Andocs G, Meggyeshazi N, Balogh L, et al. (2014) Upregulation of heat shock proteins and the promotion of damage-associated molecular pattern signals in a colorectal cancer model by modulated electrohyperthermia. Cell Stress and Chaperones 20(1):37-46, http://www.ncbi.nlm.nih.gov/pubmed/24973890

[[iii]]        Cha, J, Jeon T-W, Lee C-G, et al. (2015) Electro-hyperthermia inhibits glioma tumorigenicity through the induction of E2F1-mediated apoptosis, Int. Journal Hyperthermia, 31(7):784-792, http://www.ncbi.nlm.nih.gov/pubmed/26367194

[[iv]]        Meggyeshazi N, Andocs G, Balogh L, et al. (2014) DNA fragmentation and caspase-independent programmed cell death by modulated electrohyperthermia. Strahlenther Onkol 190:815-822, http://www.ncbi.nlm.nih.gov/pubmed/24562547

[[v]]         Andocs G, Rehman MU, Zhao Q-L, Tabuchi Y, Kanamori M, Kondo T. (2016) Comparison of biological effects of modulated electro-hyperthermia and conventional heat treatment in human lymphoma U937 cell, Cell Death Discovery (Nature Publishing Group), 2, 16039, http://www.nature.com/articles/cddiscovery201639

[[vi]]         Chi K-H. (2020) Tumour-directed immunotherapy: Clinical results of radiotherapy with modulated electro-hyperthermia, in book Challenges and solutions of oncological hyperthermia, ed. Szasz A., Ch. 12, pp.206-226, Cambridge Scholars, https://www.cambridgescholars.com/challenges-and-solutions-of-oncological-hyperthermia

Reviewer 2 Report

General comments:
The authors report on their findings on the non-temperature induced effects of radiofrequency and amplitude modulated radiofrequency fields. This is novel and first of its kind in such a controlled environment, hence the study is relevant and of interest of the journal's community. In general, methods are sufficiently well described. The results are interesting but requires some clarifications, e.g. why not all cell line experiments performed fully? Figures are low quality and hard to read. Conclusion section is almost non-existant. I recommend a more detailed conclusion section, detailing in which cell line AMRF performs better, when they are equal to RF, instead of a single speculative sentence.  Additionally, The manuscript requires a through proof-reading.

All comments:
P4L160 - define t_corr
P5L177 - Rewrite the sentence to clarify the meaning if -> as?
P5L197 - Could you clarify what additional simulation studies are required for further clarification and why it wasn't possible in this study?
P6 - How did you model the AMRF input to thermal simulations? What is the minimum step size of the thermal model?
P7L288 - It is not clear to me how you decide on the three cell lines (HT29, SW480, LoVo). I miss the explaination in the results section. Figure 3 A,B,C looks very similar for all cell lines.
P8L311 - Define AFM, P13L468 use the abbreviation or change the first one
P12L437 - What is isothermal energy? Googling the term returns nothing. Please use a more suitable term or provide a citation.
Figure 3A - Why the WB temperature go up to 44degC? Why are there no spread in apostosis rates in WB group?
P13L460 - deltaT/dtcorr correct the symbol
P14L537 - Define how the power correction has been done in the methods.
P14L543 - Why deltaT/dtcorr values are lower at the skin for AMRF group? Is it due to higher cooling effect on the skin?
P14L545 - What do you mean by "with a sign reversal"?
P14L554 - How is the non-homogeneity of the temperatures were measured? Is it similarity to skin temperature ve deep?
P15L579 - SAR has been defined previously, remove the explanation in parathesis.
P15L571 - Any hypothesis why the AM-related effects were more prominent in vivo?
P21L697 - Correct the symbol
P22L745 - What about the effect of application frequency? Do you have any comments on that?
Figure 7B - Missing

Round 2

Reviewer 2 Report

I thank the authors for their clear answers to my previously raised questions. I congratulate the authors for this very interesting study. I have no further comments.